# Nephron-Sparing Approaches in Upper Tract Urothelial Carcinoma: Current and Future Strategies

**DOI:** 10.3390/biomedicines10092223

**Published:** 2022-09-08

**Authors:** Won Sik Ham, Jee Soo Park, Won Sik Jang, Jongchan Kim

**Affiliations:** 1Department of Urology, Urological Science Institute, Yonsei University College of Medicine, Seoul 03722, Korea; 2Department of Urology, Sorokdo National Hospital, Goheung 59562, Korea; 3Department of Urology, Yongin Severance Hospital, Yonsei University Health System, Yongin 16995, Korea

**Keywords:** urothelial carcinoma, upper urinary tract, nephron-sparing, kidney function

## Abstract

Upper tract urothelial carcinoma (UTUC) is a relatively rare cancer, and much of the approach to treatment has been derived from strategies employed in treating bladder cancer. Radical nephroureterectomy (RNU) is regarded as the gold standard treatment for UTUC. However, due to potential complications, such as renal function impairment, that can affect oncologic outcomes, the demand for nephron-sparing treatment to effectively treat cancer while preserving renal function has increased. As a result, various treatment methods for low-grade, low-volume UTUC, such as segmental ureterectomy, endoscopic resection, and intraluminal therapy, have been attempted and reported. Although these treatment modalities have exhibited acceptable oncological results, further studies are required. In the future, the introduction of new technologies, such as improved diagnostic and surgical equipment, and new drug delivery systems, could enhance the effectiveness of nephron-sparing strategies in the treatment of UTUC. Additionally, understanding the biological and genetic characteristics of UTUC that distinguish it from those of bladder cancer will also aid in establishing strategies for nephron-sparing.

## 1. Introduction

Upper tract urothelial carcinoma (UTUC) originates from urothelial cells, from the ureter to the renal pelvis. UTUC is a relatively uncommon malignancy with an incidence of less than 2 per 100,000 population and accounts for approximately 5–10% of all urothelial carcinomas [1]. About 25% of UTUCs arise from the ureter, while the remaining 75% occur in the collecting system of the kidney. Radical nephroureterectomy (RNU) with bladder cuff excision, segmental ureterectomy (SU) with ureteroureterostomy or ureteral reimplantation, and endoscopic treatment have been suggested as treatment options according to tumor grade or location [2,3].

Although RNU has been accepted as the gold standard for high-risk UTUC treatment, there are several concerns. The first is perioperative complications associated with RNU. Several studies have revealed that RNU can cause approximately 8–20% of complications. As minimally invasive surgery has spread widely to reduce perioperative complications, a minimally invasive approach has also been introduced for RNU; however, it is not clear whether minimally invasive surgery reduces RNU complications [4]. Some studies reported that minimally invasive surgery reduced complications [5], whereas other studies demonstrated no difference in the complication rate between open and minimally invasive surgeries [6,7]. Campi et al. [8] reported a 44% overall complication rate after robotic RNU. Most complications were managed without interventional procedures; however, some patients required a percutaneous procedure due to bleeding or symptomatic lymphocele and reoperation due to bowel perforation. Although it is difficult to generalize because most previous studies were retrospective studies and involved diseases with relatively low incidence, the rate of these complications cannot be ignored because of the characteristics of UTUC, whereby the incidence increases with increasing age [9].

Another major complication to consider regarding RNU is the impairment of renal function. It is possible that the pre- and postoperative renal functions of patients who underwent RNU were similar to those of patients who underwent radical nephrectomy for renal cell carcinoma. However, UTUC is relatively more common in older patients, and patients with UTUC are likely to have renal insufficiency or diseases that can lead to loss of renal function, such as hypertension or diabetes mellitus [10,11]. Patients with UTUC might display reduced renal function because of ureteral obstruction due to the tumor. In a study comparing the renal function of patients who underwent radical nephrectomy and RNU, patients with UTUC were older than those with renal cell carcinoma and reported significantly reduced median estimated glomerular filtration rate (eGFR) (58.4 mL/min/1.73 m^2^ vs. 74.9 mL/min/1.73 m^2^). The postoperative eGFR was 51.3 mL/min/1.73 m^2^ in the RNU group, indicating that renal function declined compared to that before surgery. This result indicates that UTUC patients often have chronic kidney disease (CKD) at the time of diagnosis, and renal function worsens after RNU [12]. Other studies also reported a decrease in renal function of approximately 20–25% after RNU [13,14]. These studies suggest that patients with UTUC have a greater chance to develop new CKD. Although few studies investigated the effects of postoperative CKD on prognosis, preoperative renal dysfunction appears to be associated with poorer prognosis in patients with UTUC who underwent RNU [15,16]. CKD is associated with an increased risk of cardiovascular morbidity, mortality, and all-cause mortality [17]. In addition, it may be difficult for patients to receive future chemotherapy because of impaired renal function. CKD also limits a patient’s ability to receive nephrotoxic chemotherapy. For this reason, the need for nephron-sparing approaches in patients with UTUC has gradually increased over the years, and various treatment modalities are being applied in clinical practice.

## 2. Current Nephron-Sparing Strategies

### 2.1. Segmental Ureterectomy

Segmental ureterectomy (SU) can achieve more accurate staging and grading by acquiring adequate specimens while preserving renal function. It can be performed when tumor is located in the ureter; SU with ureteroureterostomy for UTUC of mid or upper ureter, distal ureterectomy with bladder cuff excision, and ureteral reimplantation for UTUC of lower ureter. Furthermore, by performing regional lymph node dissection, SU could be an option, even in patients with high-risk UTUC located in the ureter. Kim et al. [13] compared oncological and functional outcomes between RNU and SU for UTUC. They reported no difference in renal function after SU with a change in eGFR of 3.4 mL/min/1.73 m^2^; however, the change in eGFR was −15.8 mL/min/1.73 m^2^ in the RNU group. The authors also reported no difference in oncologic outcomes, even in patients with advanced-stage or high-grade tumors. Fang et al. [18] reported a significantly decreased risk of renal insufficiency in SU compared to that in RNU (mean eGFR difference = 9.32 mL/min/1.73 m^2^, *p* = 0.007). However, another study reported different results for renal function. Abrate et al. [19] compared renal function decline before and after surgery between SU and RNU in patients with preoperative eGFR ≤ 90 mL/min/1.73 m^2^. They reported that the decline in renal function was similar in both the SU and RNU groups (4.0 mL/min/1.73 m^2^ vs. 2.6 mL/min/1.73 m^2^). Based on this result, it can be inferred that the role of SU in preserving renal function might be limited in patients with considerably reduced renal function before RNU.

Studies comparing the oncological outcomes of SU and RNU have reported broadly consistent results. A previous meta-analysis showed similar intravesical recurrence-free survival (RFS) (hazard ratio [HR]: 1.35, *p* = 0.39), progression-free survival (PFS) (HR: 1.06, *p* = 0.72), cancer-specific survival (CSS) (HR: 0.90, *p* = 0.49), and overall survival (OS) (HR: 0.98, *p* = 0.93) between SU and RNU [18]. Recent studies also demonstrated that the oncological outcome of SU was not inferior to that of RNU [8,13,19].

### 2.2. Endoscopic Treatment

Endoscopic treatment is one of treatment option for UTUC with favorable characteristics. In the NCCN guidelines, favorable clinical and pathologic criteria are defined as those that satisfy the following: low-grade tumor based on cytology and biopsy, papillary architecture, tumor size less than 1.5 cm, unifocal tumor, and cross sectional imaging showing no concern for invasive disease [3]. For ureteral lesions, flexible ureteroscopy with both antegrade and retrograde approaches can be used to remove any visible tumors using laser or electrocautery. Three types of laser energy have been applied in previous studies of endoscopic ablation: holmium/yttrium-aluminum-garnet (Ho: YAG), neodymium/YAG (Nd: YAG), and thulium/YAG (Thu: YAG) lasers [20]. The Ho: YAG laser energy has a safe penetration depth because of its high absorption coefficient in water and must be used in contact with the tissue to achieve tumor ablation. Therefore, it is suitable for the treatment of superficial tumors. Nd: YAG can be used to treat bulkier tumors because of its greater penetration depth (up to 10 mm) and provides a deeper ablative effect on the tumor. The Thu: YAG laser has been used more recently than the other two types of lasers. It possesses favorable vaporization and coagulation properties for treating soft tissue disease, with a very short penetrating depth of 0.2 mm [20]. In a previous study, these lasers were used for tumor ablation either alone or in combination.

A recent retrospective study reported the long-term outcomes of 168 patients who underwent tumor ablation using retrograde ureteroscopy. The authors used Nd: YAG, Ho: YAG, or a combination of these two lasers, with a mean follow-up of 5.53 years, and reported 5-year PFS (75.2%), CSS (92.6%), and OS (80.9%). Among the 170 patients, 50 underwent RNU after a mean duration of 842 days [21]. Table 1 shows contemporary studies on endoscopic resection of UTUC. The recurrence rate varied from 19% to 90.5%, and the rate requiring RNU after ablative therapy ranged from 8.9% to 29.8%. The 5-year CSS was reported at 77.5–92.6% [21,22,23,24,25,26,27,28,29]. In a study that included patients with tumors > 2 cm, the recurrence rate was 90.3%. Although patients who underwent endoscopic treatment for palliative purposes and high-grade tumors were included, there were many differences from currently widely accepted indications, but the recurrence rate was found to be exceptionally high [25].

The percutaneous approach has great advantages in treating lesions of the renal pelvis because it enables more direct passage to the tumor. In addition, because the tumor can be more radically resected using a larger instrument, better oncological results can be expected while preserving renal function. Motamedinia et al. [24] reported the 30-year experience of 141 patients who underwent percutaneous UTUC resection. A guidewire was percutaneously inserted into the renal pelvis under ultrasound guidance, and dilation was performed using a balloon dilator. They then installed a 30 Fr nephrostomy access sheath, and a 26Fr resectoscope was used to resection tumors in the renal pelvis or calyx. The median follow-up was 66 months, 35% of the patients experienced recurrence, and 14% experienced RNU. The median RFS and median OS were 71.4 and 126 months for low-grade tumors and 36.4 and 59.6 months for high-grade tumors, respectively.

### 2.3. Intraluminal Therapy

Although endoscopic treatments might offer acceptable outcomes for selected patients with low-grade and low-volume UTUC, efforts have been made to reduce the risk of recurrence or the requirement for RNU. Several intraluminal therapies have been deployed in various studies based on the recommendation for intravesical treatment using Bacillus Calmette–Guérin (BCG) or mitomycin-c (MMC) for carcinoma in situ or high-risk non-muscle invasive bladder cancer [2,3]. Intralumial therapy can be performed using a percutaneous or retrograde ureteral catheter [30,31,32]. Percutaneous instillation was performed using a percutaneous nephrostomy catheter. The drug was injected slowly over 1–2 h while measuring the pressure in the renal pelvis [30]. Drug indwelling using a double-J stent is only possible in patients with confirmed vesicoureteral reflux, which corresponds to approximately 50% of patients. After injecting the drug into the bladder, the drug moves into the upper ureter while waiting for a certain period [31]. In patients without vesicoureteral reflux, drugs can be injected using a single-J catheter or an open-ended catheter [32].

After Orihuela et al. reported the results of BCG instillation after treatment for UTUC using Nd: YAG laser in 1988, several small studies on intralumial therapy were published [30,33,34,35,36,37]. Most of these studies used BCG and MMC, and thiotepa and BCG with IFN were used in some studies (Table 2).

The most recent study using BCG for intraluminal therapy analyzed 22 renal units that had undergone ablation for T1 or Ta UTUC [30]. BCG was injected for 2 h through a nephrostomy and administered at intervals of 1 week for 6 weeks. Thirteen renal units experienced recurrence after BCG administration. Recent studies have reported a recurrence rate of 13–75% after adjuvant BCG. They reported a recurrence rate of 40%, with a median follow-up of 42 months. MMC has also been evaluated as adjuvant intraluminal therapy. Metcalfe et al. [32] introduced data from 28 renal units receiving intraluminal therapy after endoscopic resection for Ta or T1 UTUC. They reported 3-year recurrence-free, progression-free, and nephrouretectomy-free survival rates of 60%, 80%, and 76%, respectively.

However, it is unclear whether such intraluminal therapy reduces the recurrence rate after endoscopic treatment. Rastinehad et al. [35] found that BCG after endoscopic resection or ablation did not result in significant differences in recurrence rates, regardless of the tumor grade. In low-grade UTUC, the recurrence rate was 26% for endoscopic management alone and 33% for those receiving adjuvant BCG. In high-grade UTUC, 38% recurrence was observed with endoscopic management alone and 39% with adjuvant BCG. A recent meta-analysis evaluated the impact of adjuvant intraluminal therapy on Ta-T1 UTUC. Patients who received adjuvant intraluminal treatment exhibited a similar recurrence rate to non-treated patients [38]. One reason for the limited therapeutic effect of intraluminal therapy is that the upper ureter is an unfavorable environment for such therapy. The kidneys constantly produce urine, which is excreted through the ureters into the bladder. Therefore, it may be difficult for the injected drug to be maintained at a sufficient concentration for an adequate period, which may reduce the effectiveness of adjuvant therapy.

Recently, MMC has been incorporated into gelatinous matrices to overcome the shortcomings of intraluminal therapy. UGN-101 (JELMYTO) is a reverse thermal gel that contains MMC (4 mg mitomycin per mL gel) and is a liquid at a lower temperature. It is used for the primary chemoablative treatment of low-grade UTUC. UGN-101 is instilled as a liquid through a ureteral catheter or nephrostomy tube and becomes a semi-solid gel at body temperature. It dissolves slowly in the urine over 4–6 h, resulting in prolonged contact of the MMC with the tumor. Matin et al. [39] introduced the final report of a phase 3 trial of UGN-101 for primary or recurrent biopsy-proven low-grade UTUC. Of the 71 patients who received induction of chemoablative therapy, 42 achieved a complete response, of which 41 initiated follow-up. Of these 41 patients, 56% maintained a complete response after 1 year with or without maintenance treatment. Among the various complications, ureteral stenosis was the most common treatment emergent adverse effect, and the rate of complications was higher with an increasing number of UGN-101 instillations. However, there was no statistically significant difference in the mean eGFR change before, during, or after treatment. Rosen et al. reported their initial clinical experience with UGN-101. Among the eight patients who underwent antegrade administration, 50% achieved complete remission, and 50% achieved partial remission at a median follow-up of 7 months. One patient developed a ureteral stricture requiring a ureteroscopic incision. Based on the few studies involving UGN-101, instillation of UGN-101 appears effective for primary chemoablation of low-grade UTUC, with negligible side effects.

## 3. Future Strategies for Nephron-Sparing Approaches

### 3.1. Enhancing the Accuracy of Grading and Staging of UTUC

The current indication for nephron-sparing treatment for UTUC is limited to low-grade, low-volume, noninvasive tumors. However, it is difficult to accurately distinguish the patient group with these indications through ureteroscopic biopsy and imaging studies, such as computed tomography (CT) or magnetic resonance imaging. Usually, the force used for ureteroscopic biopsy is approximately 3Fr; hence the amount of tissue that can be obtained through biopsy is very small. Therefore, in previous studies, insufficient samples were obtained for tumor staging and grading up to 30% [40]. In addition, in a study comparing the results of ureteroscopic biopsy with the pathological results of specimens from RNU or SU, approximately 30% of patients reported with low-grade tumors on biopsy were finally confirmed with high-grade cancers, and 61% of patients with T1 or less on biopsy were reported to be above T2 [41]. Imaging tests also have limitations in accurate UTUC staging. It is reported that the accuracy of distinguishing T3 or higher from T2 or lower is generally 80% or higher [42]. However, there have been few studies on the distinction between Tis, Ta, T1, and T2. Gandrup et al. [43] reported that CT has a limited role in distinguishing between T2 or higher and T1 or lower UTUC. If diagnostic technology that can perform grading and staging more accurately is developed, the nephron-sparing approach can be applied to patients with the required indications, and better oncological results may be obtained.

Another challenge in diagnosing UTUC is that it is difficult to localize tumors or carcinoma in situ (CIS) accurately using ureteroscopy. Previous studies suggested that enhanced ureteroscopy using narrow-band imaging (NBI) or photodynamic diagnosis (PDD) might help improve CIS detection. Traxer et al. [44] reviewed 27 patients who underwent standard white-light and NBI flexible ureteroscopy. They reported that NBI revealed five additional tumors in four patients and more precise borders in three tumors in three patients. Kata et al. [45] compared PDD flexible ureteroscopy and white-light ureteroscopy for the diagnosis of UTUC. They detected 48 lesions, of which 95.8% were visualized by PDD flexible ureteroscopy compared to 47.9% by conventional flexible ureteroscopy. PDD-FURS was more sensitive (95.8; range: 85.7–99.5) than WL-FURS (53.5; range: 37.7–68.8) in detecting UUT-UC. If the ability to localize tumors using ureteroscopy is increased with the development of new technology, it will help improve oncologic outcomes by not leaving residual cancer in endoscopic treatment.

### 3.2. Robot-Assisted Endoscopic Surgery for UTUC

Recently, robot-assisted endoscopic surgery has been applied in urolithiasis surgery. Tokatli et al. [46] reported the results of robot-assisted mini-endoscopic combined with intrarenal surgery for multiple stones. Forty-four renal units with complex or multiple renal stones were treated using robot-assisted mini-endoscopic combined intrarenal surgery. Retrograde access with a flexible scope was achieved using the robotic system. Endoscopy and postoperative CT confirmed that 42 renal units were stone-free following the procedure. The authors highlighted the achievement of a high stone-free rate because of its accuracy in evaluating the collecting systems. They reported grade 1 perioperative complications in three patients. In addition, endoscopic flexible ureteroscopic evaluation of the collecting system at the end of the procedure could enable the surgeon to predict the stone-free status more reliably and successfully. Although robot-assisted endoscopic surgery has not yet been used for treating UTUC, it might be applied in UTUC treatment in the future. Furthermore, this new technique, currently in use for the treatment of urolithiasis, provides excellent visualization of the calyceal and ureteral anatomy, it can be of help in removing completely the urothelial tumor.

### 3.3. Novel Drug Delivery Technologies for Intraluminal Therapy

From the perspective of intraluminal therapy, increasing the drug delivery rate using a stent may also contribute to the development of renal-sparing treatment. Barros et al. [47] introduced a new concept of drug-eluting biodegradable ureteral stents by combining hydrogel technology with conventional ureteral stents. They developed a biodegradable ureteral stent impregnated with supercritical fluid CO_2_ and four anti-cancer drugs, namely paclitaxel, epirubicin, doxorubicin, and gemcitabine. This stent can increase the contact time between the chemotherapeutic agents and the ureter as the coated drugs dissolve slowly. After drug delivery, the stent degrades without requiring a second removal procedure. The in vitro study in artificial urine solution showed a faster release in the first 72 h for the four anti-cancer drugs, after which a plateau was achieved, and finally, the stent degraded within 9 days. Microscopic evaluation of the cancer cell killing efficacy of the impregnated stent showed that the viability of cancer cells decreased by approximately 50% after 72 h of contact with the drug-loaded stents. Lim et al. [48] introduced another bilayer-swellable drug-eluting ureteral stent. This stent consists of a polyurethane-based stent spray-coated with a polymeric drug-containing layer and an expandable hydrogel layer. After the stent is inserted into the ureter, the hydrogel layer expands and contacts the urothelium; the drug which is coated in the lower layer penetrates the hydrogel layer and is delivered to the urothelium. In vitro quantification of the released drug demonstrated sustained delivery of MMC over 4 weeks. An in vivo feasibility study in a porcine model demonstrated that the swollen hydrogel coapts with the urothelium, enabling localized drug delivery to the target tissue. There were no adverse effects, such as renal function impairment, ureteral stricture, or systemic toxicity. These new drug delivery technologies are expected to increase the effectiveness of intraluminal therapy; therefore, intraluminal therapy may play a more important role in treating UTUC.

### 3.4. Understanding the Biological and Genetic Characteristics of UTUC

The discovery biomarkers that can predict tumor grade or prognosis in UTUC can help establish appropriate criteria for selecting endoscopic treatment. Mos et al. [49] reported the results of the genomic analysis of UTUC using whole-exome sequencing (WES), gene expression profiling, and protein expression analysis. WES revealed that FGFR3 mutations were observed in 74.1% of 31 UTUC samples, and FGFR3 mutations were observed more frequently in high-grade than in low-grade UTUC. RNA sequencing confirmed that the UTUC samples were divided into four subgroups that correlated with clinical variables, such as grade, stage, and recurrence. There are also studies on genes associated with the prognosis of UTUC. Audenet et al. [50] demonstrated that the risk of bladder recurrence after UTUC is significantly associated with mutations in FGFR3, KDM6A, CCND1, and TP53. Based on these genetic studies, performing endoscopic resection in patients with genetic characteristics predicted to have a favorable prognosis may lead to improved oncological outcomes.

The development of factors that predict the response to intraluminal therapy will also play an important role in developing nephron-sparing treatment. Studies related to factors that can predict the effectiveness of BCG treatment in bladder cancer have been conducted. Pichler et al. [51] performed immunohistochemical staining with PD-L1, GATA-3, IL10/IL-10 receptor, and a disintegrin and metalloproteinase (ADAM) protease from radical cystectomy specimens after BCG failure. They reported that the expression of ADAM17, which has been reported to release membrane-bound PD-L1, is high in tumor regions. Another study performed genome-wide DNA methylation analysis of NMIBCs in 26 BCG responders and 27 failures. They reported differential methylation states of six of these regions, localized in the promoters of GPR158, KLF8, C12orf42, WDR44, FLT1, and CHST11. GPR158 promoter hypermethylation was the best predictor of BCG failure, with an AUC of 0.809. Studies related to the prognosis of BCG instillation have been published in bladder cancer, but there are no studies evaluating the predictors of the therapeutic effect of intraluminal therapy in UTUC. Because UTUC has different biological and genetic characteristics from bladder cancer, it is difficult to apply the factors identified in bladder cancer to UTUC. In fact, at the time of diagnosis, 60% of UTUCs were reported to be invasive, whereas 15–25% of bladder cancers were reported to be invasive. Sfakianos et al. [52] evaluated the differences in genomic characteristics between UTUC and bladder cancer. They performed next-generation sequencing assays to identify somatic mutations and copy number alterations in 300 cancer-associated genes from the tumor and germline DNA of 83 patients with UTUC and 102 patients with bladder cancer. The authors showed that FGFR3, HRAS, and CDKN2B were more frequently mutated in UTUC than in bladder cancer, whereas TP53, RB1, and ARID1A were less frequently altered. Therefore, studies are needed to understand the biological and genetic characteristics of UTUC, and information on genetic biomarkers related to UTUC will contribute to selecting suitable patients in whom the nephron-sparing approach for UTUC can be applied and establishing treatment strategies.

## 4. Conclusions

Because UTUC has a low incidence, there are few large-scale or prospective studies related to the nephron-sparing approach. SU and endoscopic treatment to preserve renal function have shown adequate oncological results in select patients with low-volume and low-grade disease. Currently, the efficacy of intraluminal therapy is questionable. Recently, drugs using new materials to enhance drug delivery in intraluminal therapy have been developed, and some are being used in clinical practice. The role of nephron-sparing treatment will expand if new technologies are developed and applied in the diagnosis and treatment of UTUC, and the biological oncological characteristics that distinguish UTUC from bladder cancer are better understood.

## Figures and Tables

**Table 1 biomedicines-10-02223-t001:** Contemporary studies on endoscopic treatment of upper tract urothelial carcinoma.

Author	Year	Number of Patients	Approach	Ablative Energy	Median Follow Up (Months)	Upper Tract Recurrence (%)	Progression to RNU (%)	CSS	OS
Nita et al. [22]	2012	65	Retrograde: 47 Percutaneous: 18	Nd: YAG	60.0	47.7	27.7		
Vemana et al. [23]	2016	151	Retrograde/percutaneous	Ho: YAG Thu: YAG	43.0	53.0		5Y: 88%	
Motamedinia et al. [24]	2016	141	Percutaneous	Resection	66.0	35.0	14.0	NA	LG: 126 months HG: 59.6 months
Scotland et al. [25]	2018	80	Retrograde	Ho: YAG Nd: YAG	44.3	90.5	20.0	5Y 84.0%	5Y 75.0%
Musi et al. [27]	2018	42	Retrograde	Thu: YAG	26.3	19.0	9.5		
Defidio et al. [26]	2019	101	Retrograde	Ho: YAG Thu: YAG	28.7	30.7	8.9		
Bozzini et al. [28]	2020	47	Retrograde	Thu: YAG	11.7	19.2	NA		
Scotland et al. [21]	2020	168	Retrograde	Ho: YAG Nd: YAG	66.0	71.4	29.8	5Y 92.6%	5Y 80.9%
Sanguedolce et al. [30]	2021	47	Retrograde: 45 Percutaneous: 2	Ho: YAG Thu: YAG	24.0	28.3	17.0	Median 24.5 months	Median 24 months

RNU, radical nephroureterectomy; CSS, cancer-specific survival; OS, overall survival; Nd, neodymium; YAG, yttrium-aluminum-garnet; Ho, holmium; Thu, thulium; NA, not applicable; LG, low grade; HG, high grade; 5Y, 5 years.

**Table 2 biomedicines-10-02223-t002:** Concurrent studies on intraluminal therapy.

Author	Year	Agent	Purpose	Approach	No. of Renal Units	Mean Follow Up (Months)	Recurrence (%)
Kojima et al. [33]	2006	BCG	Primary therapy for CIS	Double-J stent	13	51	38
Katz et al. [34]	2007	BCG+IFN	Adjuvant therapy	Retrograde ureteral catheter	8	35	13
	2007	BCG+IFN	Primary therapy for CIS	Retrograde ureteral catheter	3	24	33
Rastinehad et al. [35]	2009	BCG	Adjuvant therapy	Antegrade nephrostomy	50	61	36
Giannarini et al. [30]	2011	BCG	Adjuvant therapy	Antegrade nephrostomy	22	41	59
	2011	BCG	Primary therapy for CIS	Antegrade nephrostomy	42	41	40
Shapiro et al. [36]	2012	BCG	Primary therapy for CIS	Retrograde ureteral catheter	11	14	18
Aboumarzouk et al. [37]	2013	MMC	Adjuvant therapy	Retrograde ureteral catheter	20	24	35
Metcalfe et al. [32]	2017	MMC	Adjuvant therapy	Antegrade nephrostomy/ Retrograde ureteral catheter	28	19	39

BCG, Bacillus Calmette–Guérin; CIS, carcinoma in situ; IFN, interferon; MMC, mitomycin C.

## Data Availability

Not applicable.

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
