# Peer review of "Nephron-Sparing Approaches in Upper Tract Urothelial Carcinoma: Current and Future Strategies"

_biomedicines, 2022, doi:10.3390/biomedicines10092223_

Round 1

Reviewer 1 Report (Previous Reviewer 2)

The authors stressed the reviewers' concerns adequately. I have no comment on this manuscript.

Author Response

Thank you very much for your kind comment.

Reviewer 2 Report (New Reviewer)

Line 76: it should be stated when is segmental ureterectomy indicated and what are the locations of the urothelial tumor in the ureter, where a segmental ureterectomy is possible. Do not forget that, for distal ureteral, low grade, small urothelial carcinomas, distal ureterectomy with a bladder cuff excision and ureteral re-implantation can be performed according to the international guidelines.

line 77: the phrase "It can be performed by regional lymph node dissection", should be re-phased to (...) a regional lymph node dissection should be performed during segmental ureterectomy"

line 243: Robot-assisted endoscopic surgery for UTUC. The paragraph can be summarized, stating that as this new technique, currently in use for the treatment of urinary lithyasis, provides excellent visualization of the caliceal and ureteral anatomy, it can be of help in removing completely the urothelial tumor.

Author Response

Line 76: it should be stated when is segmental ureterectomy indicated and what are the locations of the urothelial tumor in the ureter, where a segmental ureterectomy is possible. Do not forget that, for distal ureteral, low grade, small urothelial carcinomas, distal ureterectomy with a bladder cuff excision and ureteral re-implantation can be performed according to the international guidelines.

line 77: the phrase "It can be performed by regional lymph node dissection", should be re-phased to (...) a regional lymph node dissection should be performed during segmental ureterectomy"

Ans> I really appreciate your kind comments. I think your comment is a very important point. I added to what you mentioned. 

line 243: Robot-assisted endoscopic surgery for UTUC. The paragraph can be summarized, stating that as this new technique, currently in use for the treatment of urinary lithyasis, provides excellent visualization of the caliceal and ureteral anatomy, it can be of help in removing completely the urothelial tumor.

Ans> I think your expression is much more concise and easier to understand. ã…‘I summarized as you mentioned. Thank you.

This manuscript is a resubmission of an earlier submission. The following is a list of the peer review reports and author responses from that submission.

Round 1

Reviewer 1 Report

The current review elucidating nephron sparing approaches has been reviewed. The current review as per my view does not fit in the scope of the journal. 

The review further is devoid of any sort of illustrations which makes it difficult to follow. 

The review is more oriented towards procedures than therapies, and therefore I would recommend submitting to more surgical oncology journals. 

Reviewer 2 Report

Because there are few report related to the nephron-sparing approach for UTUC, This review is so valuable. However, this review is not well organized, as it has long been written with little relevance to nephron-sparing approach for UTUC, such as stone treatment and gene expression analysis. In addition, there are some doubts in the text that describes the contents of the cited references as shown below. 

1) Reference 9 is a very old report that evaluated patients who received RNU between 1998 and 2006. In addition, it has not evaluated the minimally invasive surgery. Therefore, the information in this report misleads the reader. Authors should use the latest report which assessed the minimally invasive surgery for RNU.

2) Reference 11 do not show the correlation between renal function and diseases such as hypertension and diabetes mellitus. Could you, please check this issue?

3) Author suggested that patients with UTUC are more likely to develop new chronic kidney disease (CKD) or worsen CKD after surgery. However, reference 12 reported that RCC patients experienced greater postsurgical declines in renal function rather than patients with UTUC. Author should check this problem.

Reviewer 3 Report

Authors reviewed one of the major concerns of UTUC. There is one minor problem.

The paper of reference 9 was too old and the mortality of this paper is too high. This study analyzed American cohort between 1988 and 2006. I recommend that you remove the sentence, including reference 9 (L45-46), because the sentence can mislead the readers.

Reviewer 4 Report

Dear Authors, 

It is an interesting and clinically useful exploration of the field. 

The authors  did not provide current guidelines on favorable clinical and pathologic criteria for nephron preservation. 

I have serious concerns regarding the chapter on influence of systemic cytotoxic? therapy in nephron sparing. The authors do not specify which systemic therapy do they refer to. 

It is written that "Systemic therapy is also a kidney-sparing treatment".  It should be more deeply clarified in what sense would systemic therapy enhance the use of nephron sparing surgical techniques.  I can not agree that the systemic therapy per se is a nephron sparing therapy.

It should be underlined that the cisplatin based  chemotherapy  is used only in patients with suitable GFR. Cisplatin is a nephrotoxic drug and can worsen the GFR.

We do not have any data that we can omit any type of surgery in case of complete remission after neoadjuvant systemic therapy.

We have no data on comparing outcomes on patients receiving neoadjuvant therapy and radiation in UTUC with neoadjuvant and surgery. 

 I think it is not clear to say...that the "effectiveness of systemic therapy in Ta/T1 patients is possibly slightly higher." DO authors mean that in this context the therapy  is effective in reducing distant relapses or reducing the size or volume of the primary? 

I would suggest to rewrite completely this chapter on systemic treatment (2.3.) or omit it from the other text.

Respectfully,